# OSA and Chronic Respiratory Disease: Mechanisms and Epidemiology

**DOI:** 10.3390/ijerph19095473

**Published:** 2022-04-30

**Authors:** Brian W. Locke, Janet J. Lee, Krishna M. Sundar

**Affiliations:** Division of Pulmonary and Critical Care Medicine, Department of Internal Medicine, University of Utah, 26 N 1900 E, Wintrobe 701, Salt Lake City, UT 84132, USA; brian.locke@hsc.utah.edu (B.W.L.); jan.lee@hsc.utah.edu (J.J.L.)

**Keywords:** sleep apnea, obstructive, obstructive pulmonary disease, lung inflammation, outcomes assessments, hypoxia, hypoventilation

## Abstract

Obstructive sleep apnea (OSA) is a highly prevalent disorder that has profound implications on the outcomes of patients with chronic lung disease. The hallmark of OSA is a collapse of the oropharynx resulting in a transient reduction in airflow, large intrathoracic pressure swings, and intermittent hypoxia and hypercapnia. The subsequent cytokine-mediated inflammatory cascade, coupled with tractional lung injury, damages the lungs and may worsen several conditions, including chronic obstructive pulmonary disease, asthma, interstitial lung disease, and pulmonary hypertension. Further complicating this is the sleep fragmentation and deterioration of sleep quality that occurs because of OSA, which can compound the fatigue and physical exhaustion often experienced by patients due to their chronic lung disease. For patients with many pulmonary disorders, the available evidence suggests that the prompt recognition and treatment of sleep-disordered breathing improves their quality of life and may also alter the course of their illness. However, more robust studies are needed to truly understand this relationship and the impacts of confounding comorbidities such as obesity and gastroesophageal reflux disease. Clinicians taking care of patients with chronic pulmonary disease should screen and treat patients for OSA, given the complex bidirectional relationship OSA has with chronic lung disease.

## 1. Introduction

Obstructive sleep apnea (OSA) is a highly prevalent disorder in the adult population affecting close to a billion adults between 30 and 69 years old [1]. The implications of the high prevalence of OSA are immense, particularly in relation to the occurrence and progression of chronic diseases. OSA occurs due to the temporary collapse of the muscular airway during sleep resulting in intermittent hypoxia and hypercapnia, sleep fragmentation, and intrathoracic pressure swings [2]. Virtually every organ system is at risk of dysfunction from chronic effects of untreated OSA, but organs within the thoracic cavity may be particularly vulnerable due to the additional consequences of wide intrathoracic pressure swings. Beyond this vulnerability of thoracic structures to OSA-related effects, the respiratory system shares a common airway and tractional effects on the upper airway from chronic lung disease may affect the propensity to upper airway collapse. Despite all these relationships, the interaction between OSA and lung disease is mainly understood as a highly prevalent comorbidity with protean effects on the lower respiratory tract.

A key challenge to understanding the consequences of OSA on chronic lung disease is that patients with OSA frequently have comorbid conditions, such as obesity, gastroesophageal reflux disease, heart failure, or metabolic syndrome, which can affect both the expression of chronic lung disease and its outcomes. Most studies utilize the apnea–hypopnea index (AHI) for OSA diagnosis and severity categorization, but there are several limitations to using AHI. The AHI does not capture hypoxic burden, OSA phenotype, and other OSA-related metrics such as sleepiness that influence the risk of poor outcomes, disease interactions, and treatment response [3]. Additionally, OSA severity based on AHI is often determined based on a single night’s study, with considerable variation occurring from the methodology used for diagnosing OSA [4,5,6]. While some studies have used specific AHI thresholds to demonstrate correlations with chronic lung disease-related outcomes, there is considerable variation amongst studies in methods used to demonstrate relationships between OSA and chronic lung disease. Beyond the demonstration of a higher prevalence of OSA with chronic lung disease, the beneficial effect of OSA treatment using positive airway pressure (PAP)-based therapies have been used for establishing causal relationships between OSA and specific lung disease.

In addition to interventional human studies, we summarize the literature on the mechanisms of disease and observational data to inform practitioners and researchers on the known impacts of OSA on airway disorders, hypoventilation syndromes, chronic cough, interstitial lung diseases, pulmonary vascular disease, and lung malignancy.

## 2. Pathophysiological Considerations

The adverse effects of OSA occur through several mechanistic pathways (Figure 1). Periodic stoppages in breathing during sleep cause intermittent decreases in airflow into the lungs and wide differences in oxygen tension between the proximal and distal airways, resulting in oxidative stress [7]. The degree of “intermittent hypoxemia” experienced by the airways is higher than any other organ system and can activate both hypoxia-driven and oxidative stress pathways [8]. The supervening oxidative stress imposed by OSA on the airways can be compounded by inflammation due to other processes such as allergic inflammation as in asthma or tobacco-induced airway damage [9]. Additionally, OSA can also affect airway immunity leading to an increased propensity for respiratory tract infection-mediated exacerbations that can progress underlying chronic airways disease [10]. Sleep fragmentation and intermittent hypoxia-driven sympathetic activation can worsen left ventricular diastolic dysfunction [11]. Elevated left heart filling pressures and sleep-related hypoxemia-mediated vasoconstriction increase pulmonary arterial pressures [12]. Perhaps most importantly, the sleep fragmentation induced by OSA leads to significant fatigue and a worsening of exercise tolerance and quality of life—symptoms already experienced by patients with chronic lung disease [13,14].

Shared risk factors between OSA and chronic lung diseases, such as obesity and GERD, may, through multiple mechanisms, lead to further deterioration of pulmonary function [14]. The relationship between OSA and many chronic lung diseases is bidirectional [9], with a worsening of one disease process leading to a deterioration of the other, as seen in COPD [14].

Two aspects of OSA-related pathophysiology that are not well understood are the effect of intermittent hypercapnia on the body and the consequences of intrathoracic pressure swings on the lungs. Although a substantial body of literature details the various effects of intermittent hypoxia on systemic and organ-specific inflammation through activation of master transcription factors such as hypoxia-inducible factor (HIF), NF-kB, AP-1, etc., and reoxygenation-induced oxidative stress [15,16], similar mechanistic effects are not known for intermittent hypercapnia. However, animal models suggest additive effects of intermittent hypercapnia on intermittent hypoxia [17]. Similarly, while researchers have postulated that OSA-mediated intrathoracic pressure changes induce recurrent tractional lung injury and fibrotic changes in idiopathic pulmonary fibrosis [18], it is unclear if such a mechanism impacts other pulmonary disorders. Further discussion of the effects of OSA on chronic lung disease will be divided into sections of airway disorders, interstitial, vascular, and other pulmonary diseases.

## 3. OSA and Asthma

Asthma is a disease of chronic airway inflammation that leads to cough, chest tightness, and dyspnea. Symptoms often worsen at night or during discrete exacerbations [19]. Asthma was historically understood to result from an allergic response mediated by T-helper 2 cells and eosinophilic inflammation. Symptom onset occurs during childhood or adolescence, and treatment with inhaled corticosteroids is generally effective. However, recent studies have demonstrated that “late-onset”, “obesity-related” and “non-allergic” asthma is common, associated with neutrophilic inflammation, and responds less favorably to inhaled corticosteroids [20]. More than half of patients with moderate to severe asthma have disease mediated by non-Th2 inflammatory cells and predominantly neutrophilic sputum [21]. Various pathophysiologic processes provoke non-allergic asthma, many of which are directly or indirectly related to OSA.

Asthma and OSA co-occur at roughly twice the expected rate [22,23], and their co-occurrence leads to difficult-to-control symptoms. The relationship between asthma and OSA is bidirectional [24]. Asthma increases the risk of subsequently developing OSA (RR 1.39 for any OSA, 2.72 for symptomatic OSA) [25]. Conversely, numerous pathophysiologic links are implicated in OSA causing or worsening asthma [26]. Apneic episodes can increase cholinergic tone, activating muscarinic receptors in the airway leading to bronchoconstriction [27]. Apneas can increase thoracic blood volume, worsening spirometric indices of airflow obstruction [28]. Acute hypoxemia itself worsens bronchial reactivity [29]. These processes lead to airway remodeling and can favor the development of neutrophilic, difficult-to-treat asthma [30].

Despite this mechanistic rationale, it has proven difficult to establish firm epidemiologic estimates of the asthma symptom burden attributable to OSA. OSA and asthma co-exist within a complex web of shared risk factors that make an assessment of the independent effects challenging [31,32]. Animal models and epidemiologic data in humans suggest that obesity increases asthma risk by various mechanisms, including endocrine signaling, mechanical effects on breathing, and direct inflammatory effects from obesity [33]. Obesity also indirectly worsens asthma control by predisposing to several conditions that aggravate asthma. For example, patients with gastroesophageal reflux disease (GERD), sinus disease, and OSA are more likely to have hard-to-control asthma [34]. Gastroesophageal reflux is known to worsen asthma control [35]. The intrathoracic pressure swings of untreated OSA may worsen reflux, particularly in patients who are obese [36]. While CPAP is known to improve GERD (regardless of whether OSA is present [37,38]), the effect of CPAP on asthma symptoms remains less clear. OSA is also associated with an increased risk of rhinosinusitis, treatment of which can lead to leads to improvement in asthma symptoms [39].

Whether associations are directly causal or indirectly related to shared comorbidities, patients with asthma and OSA experience worse asthma-related outcomes than patients with asthma alone. OSA is independently associated with 20% longer and 25% more expensive exacerbation hospitalizations when controlling for age, sex, race, ZIP code, income, hospital, comorbidity index, and obesity [40]. Several studies show a high symptom burden in patients who likely have OSA as assessed by screening instruments such as the SA-SDQ [22] or the Berlin sleep questionnaire [41]. If present, an improvement in asthma symptoms with effective treatment of OSA with CPAP could estimate the attributable effect of OSA on asthma. Several before and after OSA treatment cohort studies summarized in a meta-analysis [42] demonstrate improvement in asthma symptoms and quality of life after starting CPAP. However, synchronous management changes and regression to the mean, which is a concern because asthma symptoms are known to wax and wane, limit the validity of these findings [43]. In these studies, there has been no consistent effect on objective measures such as FEV1 or bronchial reactivity [44,45,46]. Notably, the trial done by Ng et al. [47] enrolled 101 patients with severe, inadequately controlled asthma and OSA did not show a significant difference in the primary outcome of asthma control scores (Table 1).

In summary, asthma and OSA co-occur frequently. Patients with OSA are more likely to have non-Th2 asthma that is less responsive to inhaled corticosteroids, the mainstay of asthma treatment. Further research is needed to establish how much of the burden of symptoms in these patients is due to OSA versus shared comorbidities such as GERD, sinus disease, and obesity.

## 4. OSA and Other Airway Disorders

Bronchiectasis refers to irreversible damage and dilation of the airways, leading to impaired mucociliary clearance, chronic productive cough, and increased susceptibility to pulmonary infections. Although bronchiectasis is a less common disease than asthma and COPD, it has also been associated with OSA at roughly twice the expected rate [48,49,50]. Airway collapse and reopening during intrathoracic pressure swings may perpetuate the vicious cycle thought to cause bronchiectasis [51]. Alternatively, the association may result from shared comorbidities or other confounders. Thus, further research is needed to clarify if a causal relationship exists.

Tracheobronchomalacia (TBM) results in excessive luminal narrowing of the trachea and one or both bronchi during exhalation. It has been postulated that chronic nocturnal negative pressure breathing with a closed glottis dilates the large airways [52,53]. However, while acquired TBM has high rates of co-diagnosis with OSA [54], it is unclear if OSA is indeed a risk factor for this condition [53].

## 5. OSA-COPD Overlap Syndrome

“The overlap syndrome” (OVS) originally referred to the combination of any respiratory disease and OSA [55] but now refers exclusively to the co-occurrence of COPD and OSA [56]. Despite the ambiguity of which conditions overlap, the term does convey that the resulting pathophysiology is more complex than expected if the diseases were completely distinct [57]. Both OSA and COPD are common, but epidemiologic research has not convincingly demonstrated co-occurrence beyond what is expected based on their prevalences in aggregate [58]. However, both OSA and COPD are diagnoses that represent a spectrum of disease processes. OSA occurs due to abnormal ventilatory control, abnormal pharyngeal collapsibility and anatomy, impaired upper airway dilator responses to airway collapse, decreased arousal thresholds, or some combination of these factors [59]. Similarly, COPD exists on a spectrum from pure chronic bronchitis to emphysema. The effect of OSA on COPD likely depends on which permutation of physiologic abnormalities is present in each patient with OVS, though this heterogeneity needs to be further explored [58].

While some of the physiologic changes that occur in COPD are protective against OSA, other physiologic changes increase OSA risk. Considering protective factors, tracheal traction that occurs with hyperinflation results in an inverse relationship between the amount of emphysema seen on CT of the chest and the AHI among patients referred for sleep testing [60]. In addition, the weight loss from pulmonary cachexia and decreased rapid eye movement (REM) sleep seen in patients with COPD may protect against OSA [60,61]. Conversely, patients with COPD who are heavier and have smoked longer are at increased risk of OSA due to increased upper airway inflammation [62,63]. The hypoxemia that results from emphysema and pulmonary hypertension in COPD causes a rightward shift in the oxygen–hemoglobin dissociation curve, increasing the chance that a given increase in airways resistance will cause desaturation sufficient to meet the scoring criteria for hypopneas. Lastly, sleep-related hypoventilation worsens with COPD as a result of decreased drive to breathe and skeletal muscle paralysis during REM sleep [64], leading to more sustained hypoxemia [61] and reduced sleep efficiency irrespective of any effect of OSA [65]. OSA leads to an increased drive to breathe. Some evidence suggests that patients with overlap syndrome may be protected from sleep hypoventilation compared to patients with COPD alone [66]. Accordingly, the traditional AHI-based definition of sleep apnea is unlikely to adequately encapsulate the frequency and severity of sleep breathing abnormalities when OSA occurs in severe COPD [58].

Interestingly, patients with OVS seem to have less daytime sleepiness than patients with OSA alone [61]. Therefore, standard OSA screening measures that include assessing the degree of sleepiness or fatigue may not work well in this population [67,68]. Even among patients who do not have excessive sleepiness, patients with OVS have a significantly lower health-related quality of life than patients with COPD matched for pulmonary function, gas exchange abnormalities, and comorbidities [69]. This implies that OSA leads to decreased quality of life through mechanisms other than sleepiness.

The risk of cardiovascular events and death is higher in patients with OVS than in patients with OSA or COPD alone [70]. Patients with OVS also have a higher burden of risk factors, such as hypertension, diabetes, obesity, atrial fibrillation [71], peripheral vascular disease [61], and alcohol use [61] when compared to patients with COPD alone. An analysis of 6163 patients from the Sleep Heart Health Study showed that lower FEV1 and higher AHI are associated with increased mortality risk. However, when both parameters are abnormal the risk is not increased beyond what would be expected from each abnormality alone [72]. Similarly, an administrative database review of 10,149 patients with OVS in Ontario showed that patients were at high risk for cardiovascular events, particularly with sustained hypoxemia. However, the risk was normalized in adjusted models that accounted for the comorbidity burden [70].

Direct effects from OSA may also contribute to worsened outcomes in COPD. The intermittent hypoxemia from OSA increases systemic inflammatory markers that may accelerate endothelial dysfunction and atherosclerosis [73]. More severe nocturnal hypoxemia in OSA is associated with an increased risk of cardiovascular disease, and thus the worsening gas exchange in OVS may explain some of the additive risk [3]. Airway inflammation from smoking or the chronic bronchitis phenotype of COPD could also amplify this relationship [74]. OVS patients are at increased risk of recurrent episodes of acute exacerbation of COPD (RR 1.7 compared to patients with COPD) [75] and recurrent acute hypercapnic respiratory failure [76]. OVS patients develop daytime hypercapnia failure at greater than expected rates when matched to patients with COPD and similar spirometry obstruction [77]. Patients with COPD and undiagnosed OSA have an increased risk of 30-day readmission, with a dose–response relationship (all OSA: RR 2.05; moderate OSA RR 6.68; severe OSA RR 10.01) [78]. They also experience more severe hypoxemia [79], leading to an increased risk of developing pulmonary hypertension and cor pulmonale [80]. Ultimately, this leads to increased healthcare utilization [81] and mortality [82,83] in comparison to patients with COPD alone.

Unfortunately, patients with overlap syndrome have generally been excluded from trials of treatments for OSA (or OHS, excluding FEV1/FVC < 0.7) and hypercapnic COPD (requiring COPD to be the sole cause of hypercapnia) [84]. Observational studies (summarized in Table 2) have demonstrated that patients with overlap syndrome experience improvements in spirometry, pulmonary artery pressures, blood gas parameters, and sleep architecture within three months of starting CPAP [85]. Additionally, CPAP adherence was independently associated with decreased mortality risk in 227 patients with OVS [86]. Other studies demonstrate much higher mortality in patients not treated with CPAP when compared to those who were treated, including a study of 228 patients with OVS who had a relative risk of 1.79 for mortality over ten years of follow-up [75]. Confounding by factors as a result of treatment or the healthy adherer effect remains a significant concern [87,88], as a similar methodology suggested a mortality benefit from CPAP in OSA [89], which has not been confirmed in subsequent randomized control trials [90]. It is unknown whether bilevel PAP (BPAP) or CPAP should be used, though one pilot RCT of 32 patients with OVS and chronic hypercapnia showed more effective normalization of hypercapnia over three months but no difference in lung function, cognitive function, or quality of life [84]. Further studies clarifying the treatment effects of CPAP in OVS are needed.

## 6. OSA and Hypoventilation Syndromes

The exact contribution of OSA to hypoventilation syndromes as a whole is not known and is understudied. When hypercapnia occurs in a patient who is obese and has no other pulmonary or neurologic disorders other than OSA, it is termed obesity hypoventilation (OHS) [92,93]. The pathophysiology of OHS is complex. OHS is estimated to occur in 1 in 260 US adults [92], but only a minority of patients with severe obesity have OHS [94]. Conversely, not all patients with obesity hypoventilation have severe OSA: 25% have mild or moderate OSA [95], and 10% do not have OSA at all [96]. Spirometric restriction occurs in a small portion of patients with a body mass index over 40 kg/m^2^ [97]. The factor that best differentiates OHS from normocapnic obese patients is a failure of the ventilatory control mechanisms to increase ventilation [98,99,100] in response to increased CO_2_ production resulting from larger body size [101] and higher work of breathing due to lower lung volumes [102].

Modeling and empiric data suggest that apneas, when frequent and sustained enough, can surpass the ability of the respiratory system to increase ventilation in the inter-apneic period, leading to progressive CO_2_ retention [103,104]. The bicarbonate retention induced by this nocturnal hypoventilation is thought to demarcate early or at-risk stages of OHS and may contribute to reduced ventilatory responses [104,105,106]. However, while OHS requires no other contributing conditions are present, apneas would be expected to contribute to hypercapnia to a similar or greater degree in situations where other respiratory diseases or ventilatory control abnormalities limit the maximal inter-apneic ventilation.

Three case series investigating the prevalence of sleep apnea among patients admitted to various ICUs with hypercapnia have all found rates of severe OSA (AHI > 30 events/h) above 50% [76,107,108], which is several times higher than would be expected in the population matched for age and BMI [109].

Neural and neuromuscular diseases often develop upper airway dysfunction and central respiratory control abnormalities leading to obstructive apneas, which further increases the risk of hypercapnic respiratory failure [110]. Additionally, OSA is known to increase the risk of opiate-induced respiratory depression (OR 1.4) [111], potentially related to the prolonged apneas that occur due to an unstable ventilatory response [112,113].

Patients with hypercapnia experience a very high burden of disease, though the fractional contribution of OSA is unclear. CPAP treatment is frequently effective for patients with OHS and other causes of hypoventilation, though it has effects on the respiratory system independent of abolishing OSA [98]. Admissions with hypercapnia are very common [114], occurring approximately as often as admissions for pulmonary embolism [115]. Data suggest that patients with any cause of hypercapnia have high healthcare utilization [116], morbidity [117], and mortality [118,119]. Diagnosis of hypoventilation is often delayed or missed [120], and patients experience increased healthcare utilization leading up to diagnosis than matched controls [121]. Observational data primarily focusing on obesity hypoventilation suggest that patients with hypoventilation syndromes in the hospital should be empirically started on PAP therapy to reduce mortality (4.9% vs. 22.7% at six months) and risk of readmission [122,123,124].

## 7. OSA and Refractory Chronic Cough

Unexplained or refractory chronic cough is an important problem seen in both primary care and specialty clinics that results in significant healthcare utilization [125]. Patients with chronic cough undergo extensive testing and treatment for GERD, upper airway cough syndrome, and cough-variant asthma with varying improvements in cough resolution [125]. Following the finding of a retrospective study demonstrating the benefit of CPAP on cough improvement, a number of studies have shown a beneficial effect of treatment of comorbid OSA with CPAP on cough resolution [126,127,128]. While improvement in GERD due to CPAP has been postulated as the reason for the improvement in cough, recent understanding of chronic cough as occurring due to cough hypersensitivity has led to the need to understand CPAP benefit on the neuropathic bases of chronic cough [129]. Besides reduction in GERD [130], CPAP therapy also reduces mechanical trauma engendered to the upper airway during hypopnea–apnea events and causes lung inflation that may modulate the cough reflex [131].

## 8. OSA and Interstitial Lung Disease

Despite the variability in reported frequency of OSA among patients with interstitial lung disease (ILD), its prevalence seems to be disproportionately high, even when compared against an age- or BMI-matched population without ILD [109]. Prospective studies on patients with newly diagnosed idiopathic pulmonary fibrosis (IPF), the most common idiopathic form of ILD, report a prevalence of any sleep apnea (defined as AHI ≥ 5 events/h) between 59 and 89% and moderate to severe sleep apnea (defined as AHI ≥ 15/h) between 15 and 68% [132,133,134,135]. A few studies that include patients with other forms of ILD, such as scleroderma, hypersensitivity pneumonitis, and connective tissue disease-related ILD, have similarly found an increased prevalence of sleep-disordered breathing, though it is not known if particular forms or patterns of ILD increase the risk more than others [136,137,138]. A recent meta-analysis identified a 61% prevalence of OSA among patients with various forms of ILD, with 26% having moderate to severe disease [139]. Of note, despite the high frequency of OSA among patients with ILD, this condition remains underdiagnosed [140].

The pathophysiologic relationship between OSA and ILD appears to be bidirectional. Untreated OSA, through mechanisms including repetitive tractional alveolar injury, intermittent hypoxia, and nocturnal reflux, may expedite fibrotic lung injury and thereby worsen clinical outcomes [18]. On the other hand, restrictive pulmonary physiology in ILD reduces tracheal traction and may result in an increased propensity to oropharyngeal collapse [141,142], the hallmark of OSA. Although obesity is an important risk factor for OSA in the general population, its impact on the presence and severity of OSA among patients with ILD is less clear, with many studies showing comparable BMIs between ILD patients with and without OSA [133,135].

Among patients with ILD, OSA is not only associated with worse sleep quality and quality of life [143], but also potentially worse outcomes with respect to mortality and progression of the disease [144,145]. In a large administrative database from Ontario, the risk of respiratory-related hospitalization and all-cause mortality was reduced in IPF patients who had received polysomnography compared to those who did not [146]. While a retrospective multicenter study on patients with ILD and moderate to severe OSA did not find that CPAP impacted mortality or progression-free survival [147], at least three prospective trials that limited enrollment to patients with a new diagnosis of IPF and moderate to severe OSA have found consistent improvements in measures of sleep-related quality of life and potentially mortality, particularly among patients who are compliant with therapy (Table 3) [135,148,149]. More prospective trials on patients with various forms of ILD are needed.

Unfortunately, despite the potential benefits of CPAP therapy in these patients, there are specific challenges in this population that can make adherence to treatment more challenging, including, but not limited to, the presence of a chronic cough, comorbid anxiety or depression, and concomitant use of steroids, which can disturb sleep and also worsen the severity of OSA. Furthermore, the progressive nature of many forms of ILD, particularly IPF, may result in the need to adjust pressures and modes and also lead to difficulty with tolerating standard PAP interfaces, highlighting the need to diagnose OSA early in the course of the disease to maximize tolerance and potential benefits of treatment.

## 9. OSA and Sarcoidosis

Sarcoidosis is a granulomatous disorder of unknown etiology that impacts a wide range of organ systems, most commonly the lungs. OSA frequently occurs in patients with sarcoidosis and may be the result of granulomatous inflammation of the upper airway tract (involved in 5% of sarcoidosis cases [150]), weight gain related to corticosteroid treatment, or sarcoid neuropathy [151]. OSA is more severe in patients with pulmonary involvement in their sarcoidosis [152]. It is not known if the systemic inflammation from OSA contributes to sarcoidosis disease activity [151].

Sleep disturbances occur roughly three times more frequently in patients with sarcoidosis than in matched controls [153]. OSA is a strong independent predictor of excessive daytime somnolence in this population [154]. However, fatigue is one of the principal constitutional symptoms of sarcoidosis, occurring in roughly 60% of patients [155]. Despite being similarly common in patients with OSA [156], OSA is not an independent predictor of excess fatigue in one large survey [154]. On the other hand, in the 3 months after starting CPAP, patients with sarcoidosis and OSA had improvements in both fatigue and sleepiness [152]. The lack of a control group limits inferences, and further investigation is needed to determine if the relationship is causal.

Pulmonary hypertension occurs in 5–20% of patients with sarcoidosis [151], often directly due to disease activity. Therefore, it is recommended that patients with pulmonary hypertension undergo evaluation and treatment of sleep apnea to avoid additive effects from untreated OSA on their pulmonary vascular disease [151].

## 10. OSA and Pulmonary Vascular Disease and Venous Thromboembolism

Significant impacts on pulmonary hemodynamics have been observed in patients with sleep apnea. During obstructive events where the diaphragm contracts against a closed glottis, there are large swings in intrathoracic pressure translating into an increase in transmural pulmonary artery pressure, which can be further exacerbated by hypoxemia, which drives pulmonary arterial vasoconstriction [157]. Despite these acute hemodynamic effects, OSA alone without concomitant respiratory or cardiac disorders is not a significant risk factor for chronic pulmonary hypertension. However, this is not true in OSA patients with concurrent obesity hypoventilation syndrome and COPD, where hemodynamics does seem to be impacted by nocturnal hypoxemia [158,159,160].

Mechanistic data support that OSA may predispose to venous thromboembolism, but epidemiologic data have thus far been mixed. The oxidative stresses generated by intermittent hypoxia and the sleep fragmentation due to untreated OSA lead to increased hematocrit, coagulation factors, platelet activity, and impaired fibrinolytic activity [161]. Interventional trials have shown variable effects of CPAP in ameliorating these abnormalities [161]). Unfortunately, obesity, age, and sedentary behavior are shared risk factors that could lead to correlation without a causal relationship. The single highest-quality retrospective cohort found that patients with OSA are at higher risk for VTE but that the effect was not independent of BMI [162]. A systematic review of 18 studies [163] documented a 2- to 4-fold increase in venous thromboembolism across a broad range of populations and methodologies, including peri-operative patients, prospective cohorts of patients at increased risk, and population-based retrospective cohorts. However, the included studies had a less robust confounder adjustment. No confirmatory evidence showing the ability of CPAP to mitigate VTE risk is available.

## 11. OSA and Lung Cancer

An analysis of the Wisconsin Sleep Cohort [164] with a 22-year follow-up showed a dose–response relationship between OSA severity by both AHI (HR 4.8 for severe) and nocturnal hypoxemia (HR 8.6 for severe) and cancer mortality after adjustment for age, sex, BMI, and smoking. A subsequent analysis of 5000 patients in Spain [165] found that hypoxemia severity, particularly in male patients under age 65, conferred an independent risk of incident lung cancer.

Several causal mechanisms may explain the relationship reviewed in detail by Hunyor et al. [166]). Animal models suggest that hypoxic upregulation of HIF, NF-KB, and Wnt-signaling may influence tumorigenesis and metastasis risk [167,168]. Additionally, sleep disruption increases sympathetic inflammation and immune dysregulation, which may encourage carcinogenesis.

A recent systematic review and meta-analysis summarizing 4.8 million patients in four studies concluded that OSA conferred a roughly 25% increased hazard of incident lung cancer [169]. However, significant heterogeneity and the difficulty in adequately controlling for confounders limit the strength of this assertion. Contrary to expectation, the only included study to examine the risk of metastasis and mortality found no increase in risk after controlling for confounders [170]. More evidence is needed before confident assertions can be made about the influence of OSA on lung cancer.

## 12. Future Directions for Research

Given the high prevalence of OSA and its overlap with chronic lung disease, particularly COPD, research priorities in overlap syndrome have recently been outlined [58]. Major needs in OSA-COPD overlap syndrome include research to characterize further the nature of sleep-related ventilatory disturbances and their mechanistic bases. Epidemiologic characterization of the relationship between COPD severity and OSA, and which patient/disease characteristics modify the excess risk of lung disease progression, and other adverse outcomes, would greatly improve observational and randomized studies [58]. A similar need to understand epidemiological and pathophysiological relationships exists for other pulmonary overlap disorders such as interstitial lung disease [171] and pulmonary hypertension [12]. There is a pressing need to understand mechanistic pathways by which sleep-disordered breathing worsens lung disease, because there may be shared inflammatory and fibrotic pathways that may improve with the treatment of OSA. The increased risk of adverse outcomes from respiratory infections in untreated OSA has been shown in COVID-19 [172,173,174], which highlights the need to understand the effects of untreated OSA on innate immunity. The implications for reducing the burden of respiratory disease through reducing acute exacerbations of chronic disease and lower respiratory infections by treatment of comorbid OSA are huge. OSA-driven airflow changes and downstream effects represent a key component of the endogenous exposome that the respiratory tract endures during the lifetime [175], and large-scale epidemiologic data derived from electronic medical records have the potential to clarify the role of OSA treatment in reducing respiratory disease-related morbidity and mortality

## 13. Conclusions

While OSA impacts virtually all organ systems, its downstream effects on the lungs are not unexpected since a “unified airway” can amplify OSA’s effects through exaggerated oxygen tension and intrathoracic pressure swings. Despite the extensive literature on the overlap of OSA with almost all types of chronic lung diseases (COPD, asthma, chronic cough, interstitial lung disease, pulmonary hypertension, and sarcoidosis), there is inadequate recognition and treatment of comorbid sleep-disordered breathing by practitioners encountering chronic lung disease. In addition to directly addressing the adverse effects of OSA, CPAP improves upper airway and esophageal function, lessening aspiration and GERD, respectively, providing added benefits beyond what is obtained through improving functional residual capacity and gas-exchange. Despite the incomplete understanding of the relationship between OSA and various chronic lung diseases, the treatment of comorbid sleep-related breathing problems remains a promising avenue for improving not only quality of life, but also, potentially, morbidity and mortality in these patients [75,122,149].

## Figures and Tables

**Figure 1 ijerph-19-05473-f001:**
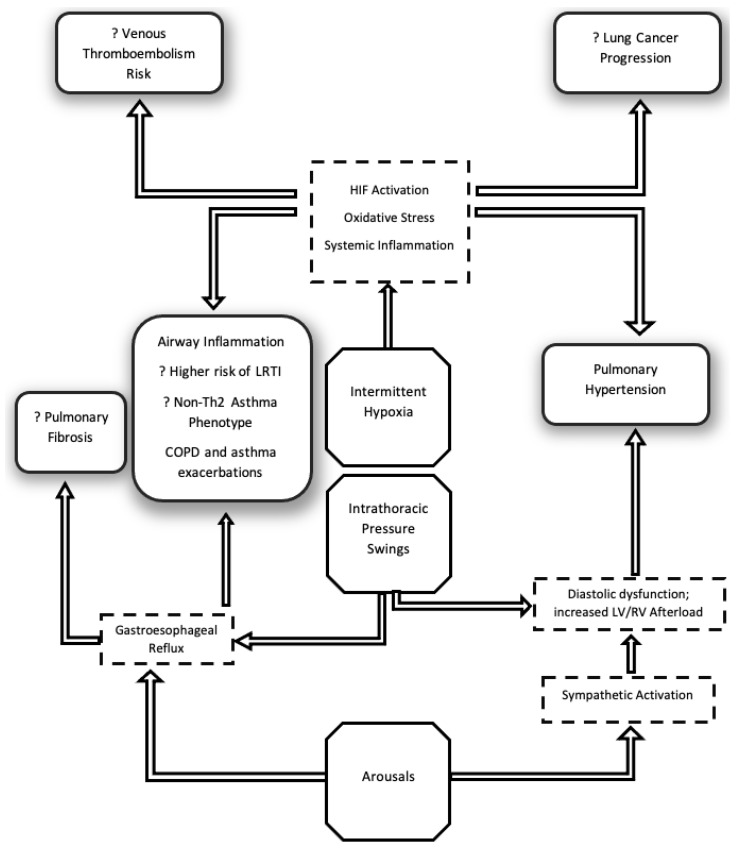
Pathophysiologic Consequences of Obstructive sleep apnea (OSA): A schematic representation of OSA-driven pathways leading to chronic lung disease. OSA leads to adverse effects through at least three causal pathways (represented by octagons): intermittent hypoxia, intrathoracic pressure swings, and sleep fragmentation (arousals). These lead to several pathophysiologic changes (represented by dashed squares) that may cause clinical events (rounded boxes). Abbreviations: LRTI—Lower respiratory tract infection; HIF—Hypoxia-inducible factor; LV/RV—Left ventricle/Right ventricle; Non-Th2 Asthma Phenotype—A type of asthma characterized by non-allergic, neutrophilic inflammation.

**Table 1 ijerph-19-05473-t001:** Key studies in the treatment of Asthma–OSA overlap.

Study	Design	Patients	Key Findings	Limitations
Ng et al. (2018) [47]	RCT, CPAP vs. no CPAP. 122 patients tested, 37 randomized. Hong Kong. ACT	Adults with asthma who snore and have nocturnal symptoms. Randomized if PSG showed AHI > 10 events/h	33.6% had AHI over 10 events/hr. No difference in ACT score change (CPAP + 3.2, Control + 2.4, *p* = 0.57)	Small sample size.
Davies et al. 2018 [42]	Metanalysis of 8 observational studies. Mean duration of CPAP use 19.5 weeks.	Adults with Asthma and OSA treated with CPAP	ACQ scores improved (0.59, 2 studies). No difference in FEV1 (4 studies)	High risk of bias, significant heterogeneity.
Serrano-Pariente et al. (2017) [46]	Before–after; 6-month follow-up; 99 patients	Asthma + new diagnosis of OSA starting CPAP	ACQ improved from 1.39 to 1.0	No control group, unclear if the effect in addition to regression to mean. Mean change of 0.39 less than MCID (0.5)

Abbreviations: RCT = randomized controlled trial, CPAP = continuous positive airway pressure, ACT = asthma control test, a survey instrument to assess asthma control, PSG = polysomnography, AHI = apnea hypopnea index. AQLQ = Asthma quality of life questionnaire, MCID = minimal clinically important difference.

**Table 2 ijerph-19-05473-t002:** Key studies in the treatment of COPD–OSA overlap.

Study	Design	Patients	Key Findings	Limitations
Marin et al. (2010) [75]	Prospective cohort study, *n* = 651, 9.4 years mean follow-up, Sleep clinics in Spain.	Patients with COPD referred for sleep evaluation. Compared COPD, OVS and declined CPAP, OVS and tried CPAP	COPD and OVS with CPAP had similar mortality, but RR for AECOPD (1.7) and death (1.79) in OVS without CPAP were elevated.	Acceptance of CPAP recommendations is likely a marker for a broad range of health behaviors that influence mortality.
Machado et al. (2010) [91]	Prospective cohort study, *n* = 95. Pulmonary clinics in Brazil. Median follow-up 41 months. Cox proportional hazard modeling.	Patients with COPD on LTOT for 6+ months with mod–severe OSA on PSG. Compared patients who started CPAP to those who didn’t.	HR for death was 0.19 in the CPAP-treated group.	Acceptance and insurance coverage of CPAP reflects health behaviors and socioeconomic status.
Stanchina et al. (2013) [86]	Retrospective cohort. *n* = 227 patients with overlap	Diagnosis by ICD code and confirmed by survey. Associated CPAP use in first 3 months to mortality.	Each hour of nightly adherence is associated with an HR of 0.71 for mortality. Age is also independently associated (HR 1.14), but FEV1, smoking, and O_2_ are not.	Retrospective, no control group. CPAP benefit is associated with strong confounders for mortality [87]
Toraldo et al. (2010) [85]	Prospective case series, *n* = 12. Italy. Follow-up at 3, 12, and 24 months	Patients with BMI 30+, moderate obstruction, severe OSA. Starting nasal CPAP	FEV1, FRC, mPAP, PaCO_2_, and PaO_2_ improved by 3 months, then stable. ESS improved at 3 and further at 12 months.	No control group. Severe disease (majority hypercapnic at the start)

Abbreviations: RR = relative risk, HR = hazard ratio, COPD = chronic obstructive pulmonary disease, OVS = COPD-OSA overlap syndrome, LTOT = long term oxygen therapy, CPAP = continuous positive airway pressure, PSG = polysomnography, FEV1 = forced expiratory volume in 1 s, ICD = International Classification of Disease, BMI = body mass index, AECOPD = acute exacerbation of COPD, RR = relative risk, HR = hazard ratio, ESS = Epworth Sleepiness Scale, FRC = functional residual capacity, mPAP = mean pulmonary artery pressure.

**Table 3 ijerph-19-05473-t003:** Key studies in the treatment of OSA-ILD.

Study	Design	Patients	Key Findings	Limitations
Mermigkis et al. (2013) [149]	Prospective single-center cohort study *n* = 23 (Greece)	Patients with incident IPF underwent a Type I PSG; 12 patients were found to have moderate to severe OSA * and were placed on CPAP.	With CPAP, there was a significant improvement in sleep-related QOL, as measured by the FOSQ at 1,3, and 6 months. There was no significant change in other QOL measures, including ESS.	Single-center, small population. Compliance was poor, with 2/12 patients not able to tolerate PAP therapy.
Mermigkis et al. (2015) [148]	Prospective multicenter cohort study *n* = 92 (Greece)	Patients with incident IPF underwent a Type 1 PSG; 60 patients had moderate to severe OSA *; of these patients, 55 agreed to CPAP.	Good compliance group ** (*n* = 37) had significant improvements in all QOL measures at one year. At two-year follow-up, significant mortality benefit with good compliance, 3 deaths vs. 0 deaths.	Poor compliance.
Adegunsoye et al. (2020) [147]	Retrospective observational multicenter cohort study (United States) *n* = 160	Patients with ILD who had undergone a Type 1 PSG; 94 patients with moderate to severe OSA *; of these patients, 51 with untreated/poor compliance, and 43 with good compliance **	No difference in all-cause mortality, progression-free survival, or lung transplantation with moderate/severe OSA treatment. Among sub-population requiring supplemental oxygen, CPAP compliance was associated with improved progression-free survival.	Retrospective study. Inclusion of various forms of ILD, including IPF. Unclear length of ILD diagnosis. Low overall event rate led to the study being underpowered.
Papadogiannis et al. (2021) [135]	Prospective single-center cohort study *n* = 45 (Greece)	Patients with incident IPF underwent Type 1 PSG; 29 patients with moderate to severe OSA were started on CPAP.	Of the 29 patients on CPAP, 11 (38%) had good compliance **. Compared to poor compliance, good compliance group saw improvements in QOL measures before and after CPAP. No survival benefit was seen, except in a sub-group averaging ≥ 6 h usage, 70% of nights.	Poor compliance.

Abbreviations: FOSQ = Functional Outcomes of Sleep Questionnaire; ESS = Epworth Sleepiness Scale; ILD = interstitial lung disease; IPF = idiopathic pulmonary fibrosis, * Moderate to severe OSA defined as AHI ≥ 15/h, ** Good compliance is defined as ≥70% of days with ≥4 h of usage.

## Data Availability

This review does not report any data.

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
