# Peer review of "OSA and Chronic Respiratory Disease: Mechanisms and Epidemiology"

_ijerph, 2022, doi:10.3390/ijerph19095473_

Round 1

Reviewer 1 Report

This article reviews the relationship between OSA and chronic lung diseases. The aim of the article is of high interest. It is certainly difficult from the known epidemiology to estimate if sleep apnea is just a comorbid disorder or if it is part of the chronic lung disease. The introduction should shed more light on this difficult point since it is problematic to understand the causal relation. This is even more important considering that many of the cited studies report patients with an AHI >5/h without any range, which after adjusting for age could be a normal finding. Other confounders i.e. obesity, diabetes etc. could find more consideration.
The authors should clarify if this is a systematic review. If so, they should include their literature search mode. 
Figure 1 is difficult to understand: The entry box shows airway disease and worsening of asthma, COPD etc. What is meant by airway disease? Do authors mean any kind of airway disease or sleep apnea? Why does this box contain worsening of asthma etc. which would be a consequence and should be in another box? Where does sleep apnea come in?
Line 211: Table 2 shows more prospective than retrospective studies
The conclusion sums up the difficulties encountered reviewing the relationssip between OSA and chronic lung disease. Here the authors make clear that their intention is to pinpoint the underdiagnosis of OSA in diagnosis and treatment of lung diseases. This could be explained in more detail in the introduction.

Author Response

Reviewer 1: 

“This article reviews the relationship between OSA and chronic lung diseases. The aim of the article is of high interest. It is certainly difficult from the known epidemiology to estimate if sleep apnea is just a comorbid disorder or if it is part of the chronic lung disease. The introduction should shed more light on this difficult point since it is problematic to understand the causal relation.”

The introduction has been expanded to better frame the goals of the manuscript and to address the question of whether OSA is a comorbid disorder or part of the chronic lung disease. Included in this is a discussion of the limitations of prior research, scope of the review, and an explanation of the impetus for summarizing this field.

“This is even more important considering that many of the cited studies report patients with an AHI >5/h without any range, which after adjusting for age could be a normal finding. Other confounders i.e. obesity, diabetes etc. could find more consideration.”

Variable definitions of OSA (including AHI > 5; vs AHI > 15; with or without symptoms) are a limitation to the existing research on obstructive sleep apnea, as the risk of adverse consequences varies on these spectrums. We have tried to note which definition is used (e.g. ILD section) where possible – though we have included studies that use each of the above thresholds. Additionally, we have included more discussion of this in the introduction

“The authors should clarify if this is a systematic review. If so, they should include their literature search mode.“

This manuscript is better termed as a narrative review than a systematic review. Given the large number of topics covered and variable size and scope of the literature, the search strategies needed to vary for each topic. In general:

  1. Author knowledge of impactful research was used to identify key studies
  2. Guidelines (when present) and contemporary reviews were searched and referenced articles were evaluated for inclusion
  3. MESH search terms of [Obstructive Sleep Apnea] + [Lung Disease] were used in Medline, Scopus, and Google scholar to identify less well known research.

“Figure 1 is difficult to understand: The entry box shows airway disease and worsening of asthma, COPD etc. What is meant by airway disease? Do authors mean any kind of airway disease or sleep apnea? Why does this box contain worsening of asthma etc. which would be a consequence and should be in another box? Where does sleep apnea come in?”

Figure 1 has been updated. Specifically, Airway disease has been deconstructed into component entities (airway inflammation when discussing pathophysiology; Asthma or COPD when discussing the disease entity). Additionally, additional wording has been added to clarify the intent of the different boxes.

“Line 211: Table 2 shows more prospective than retrospective studies”

Changed to “observational”

“The conclusion sums up the difficulties encountered reviewing the relationship between OSA and chronic lung disease. Here the authors make clear that their intention is to pinpoint the underdiagnosis of OSA in diagnosis and treatment of lung diseases. This could be explained in more detail in the introduction.“

More discussion of this has been added to the introduction (summarized in response to earlier suggestion), as well as to the new “Future Directions” section suggested below.

Reviewer 2 Report

The manuscript-review titled "OSA and Chronic Respiratory Disease: Mechanisms and Epidemiology" (ijerph-1684484) submitted to IJERPH, describes a very interesting and a the same time very narrow, but important topic related to OSA and Chronic Respiratory Disease.

Review is quite well organized and written. Manuscript is divided properly and after simple abstract, authors wrote short introduction with perfectly ilustrated and explained Figure 1  and after that authors decided desrcibed OSA with common disorder separately in few chapters.

Manuscript provided most important information related to OSA, everything is easy to read, authors put a lot of effort to prepare very readable figures and tables.

To summarize review is based on almost 160 references, concluded that OSA impacts virtually all organ systems with almost all types of chronic lung diseases (COPD, asthma, chronic 392 cough, interstitial lung disease, pulmonary hypertension, and sarcoidosis), there is still limit of recognition and treatment of comorbid sleep-disordered breathing by practitioners encountering chronic lung disease.

Author Response

Reviewer 2. 

“The manuscript-review titled “OSA and Chronic Respiratory Disease: Mechanisms and Epidemiology” (ijerph-1684484) submitted to IJERPH, describes a very interesting and a the same time very narrow, but important topic related to OSA and Chronic Respiratory Disease.

Review is quite well organized and written. Manuscript is divided properly and after simple abstract, authors wrote short introduction with perfectly illustrated and explained Figure 1  and after that authors decided described OSA with common disorder separately in few chapters.

Manuscript provided most important information related to OSA, everything is easy to read, authors put a lot of effort to prepare very readable figures and tables.

To summarize review is based on almost 160 references, concluded that OSA impacts virtually all organ systems with almost all types of chronic lung diseases (COPD, asthma, chronic 392 cough, interstitial lung disease, pulmonary hypertension, and sarcoidosis), there is still limit of recognition and treatment of comorbid sleep-disordered breathing by practitioners encountering chronic lung disease.“

We thank the reviewer for their kind comments.

Reviewer 3 Report

The review proposed by Brian W Locke and coll. is a very interesting point of view and highlights the importance of OSA in the spectrum of chronic lung disease. In particular it focuses the attention on some chronic diseases not immediately and commonly linked/associated to OSA; in some of them is very hard to distinguish, from the evidence so far, the role of  OSA, whether a factor involved in the pathogenesis of chronic lung diseases or simply a “guilty by-stander”. The review is well written and documented.

In my opinion some points are worth of discussion and, if authors agree, of improvement of the paper.

Firstly, it is not so clear to me the scope of the review; is it to better get inside possible mechanisms of interrelationships between OSA and chronic lung disease (as it seems looking at figure 1) or just to “call” for new studies and evaluate impact of OSA? Better define at the beginning.

Concerning the role of OSA in asthma: reading this section I couldn’t avoid to think that asthma is a disease of young people, and also that obesity and OSA is a matter of concern in childhood; would you comment on that?

Line 161 in COPD section: It is also true the opposite: some of the predisposing factors to OSAS (obesity) is protective against increased air trapping in the lungs and delayed occurrence of hyperinflation.

Concerning this point, issue mentioned in line 200 is crucial: COPD is a great container in which   emphysema patients and chronic bronchitis are opposite edges of the whole phenotyping of the disease. It would be worthy mentioning the possibility (in future) to better characterize overlap syndrome as OSA + phenotype of COPD?  Are there any evidences in literature concerning association of OSA and specific phenotypes of COPD (i.e. emphysema)?

Line 213-214: what about the argue that bilevel ventilation might be a better therapeutic option than CPAP in these patients?

OSA and chronic cough: compared to other sections, this part is a bit too concise.

OSA and chronic Lung diseases. Likewise COPD, ILD is a very complex and variegate issue: Is there any information about incidence of OSA and different patterns of ILD? What about connective tissue associated interstitial lung disease and OSAS?

Line 238: Do you mean mild sleep apnea

OSA and hypoventilation syndromes: I would consider to move this section immediately after OSA and overlap syndromes.

I’m not so sure that this comment be correct, but I couldn’t help to think about the evidence of the occurrence of OSA in sever COVID-19, and consequently its role in the occurrence of long sequalae of SARS-COV2-related ARDS. Do you think that this is worthy of a final comment?

Might Figure 1 be improved in quality and “appeal”?

Finally, I would consider to separate Conclusions of the evidence of literature, from a section of  “ Directions for further research”  

Author Response

Reviewer 3

“The review proposed by Brian W Locke and coll. Is a very interesting point of view and highlights the importance of OSA in the spectrum of chronic lung disease. In particular it focuses the attention on some chronic diseases not immediately and commonly linked/associated to OSA; in some of them is very hard to distinguish, from the evidence so far, the role of  OSA, whether a factor involved in the pathogenesis of chronic lung diseases or simply a “guilty by-stander”. The review is well written and documented.

In my opinion some points are worth of discussion and, if authors agree, of improvement of the paper.”

“Firstly, it is not so clear to me the scope of the review; is it to better get inside possible mechanisms of interrelationships between OSA and chronic lung disease (as it seems looking at figure 1) or just to “call” for new studies and evaluate impact of OSA? Better define at the beginning.”

An additional paragraph has been added to the introduction to more thoroughly explain the scope, methodology, and goals of the project. In sum, we seek to summarize evidence from mechanistic studies, epidemiologic research, and (where available) interventional trails to inform both current practice and future research. Accordingly, as suggested below, an additional section of future research directions has been added.

“Concerning the role of OSA in asthma: reading this section I couldn’t avoid to think that asthma is a disease of young people, and also that obesity and OSA is a matter of concern in childhood; would you comment on that?”

An additional point has been added to this section clarifying that obesity also contributes to the causal web of contributing conditions that must be accounted to estimate direct effects of OSA on asthma.

“Line 161 in COPD section: It is also true the opposite: some of the predisposing factors to OSAS (obesity) is protective against increased air trapping in the lungs and delayed occurrence of hyperinflation.”

“Concerning this point, issue mentioned in line 200 is crucial: COPD is a great container in which   emphysema patients and chronic bronchitis are opposite edges of the whole phenotyping of the disease. It would be worthy mentioning the possibility (in future) to better characterize overlap syndrome as OSA + phenotype of COPD?  Are there any evidences in literature concerning association of OSA and specific phenotypes of COPD (i.e. emphysema)?”

Both the above comments are addressed together.

A discussion of the myriad pathophysiologic phenotypes of both OSA and COPD has been added, as this likely modifies the effect of each disease on the other.

Additionally, data from the COPD-Gene study (https://www.ncbi.nlm.nih.gov/pmc/articles/PMC5015748/) does suggest that emphysema is ‘protective’ (decreased AHI) in OVS when other characteristics are adjusted for. This was included, but the discussion of this reference has been expanded to emphasize this point (starting on line 341)

“Line 213-214: what about the argue that bilevel ventilation might be a better therapeutic option than CPAP in these patients?”

One small RCT has been performed on patients with the overlap syndrome and hypercapnia, showing an improvement in 3 month PaCO2 but no difference in functioning, which has been added to the discussion to help clarify this point. https://jcsm.aasm.org/doi/10.5664/jcsm.9506

“OSA and chronic cough: compared to other sections, this part is a bit too concise.”

This section has been expanded

“OSA and chronic Lung diseases. Likewise COPD, ILD is a very complex and variegate issue: Is there any information about incidence of OSA and different patterns of ILD? What about connective tissue associated interstitial lung disease and OSAS?”

Thank you for the comment. Since IPF is the most common idiopathic ILD and best studied, this was the main focus of this limited review. Studies evaluating OSA in other ILD populations are somewhat limited, though a meta-analysis published in 2021 by Cheng et al did include in their analysis other studies (inclusive of connective tissue disease associated ILD). Though good data does not exist on whether specific patterns increase the risk (the aforementioned meta-analysis does not contain information about the number of patients with the underlying risk, and thus the conditional probability of OSA given the presence of a type of ILD cannot be calculated). A sentence clarifying this point was added.

“Line 238: Do you mean mild sleep apnea”

Added wording to clarify that any sleep apnea (mild, moderate, or severe) included in this

“OSA and hypoventilation syndromes: I would consider to move this section immediately after OSA and overlap syndromes.”

Agreed, this change has been made and section numbers have been updated.  (This change was not tracked, to avoid making the document harder to read)

“I’m not so sure that this comment be correct, but I couldn’t help to think about the evidence of the occurrence of OSA in sever COVID-19, and consequently its role in the occurrence of long sequalae of SARS-COV2-related ARDS. Do you think that this is worthy of a final comment?”

While there is some evidence linking both the susceptibility and severity of COVID to OSA, most of this is based on retrospective correlative studies and confounded by other associations particularly obesity. We have however included mention of this association between COVID and OSA in our conclusions.

‘Might Figure 1 be improved in quality and “appeal”?’

Agree – the aesthetics of this figure have been improved in the latest revision.

“Finally, I would consider to separate Conclusions of the evidence of literature, from a section of  “ Directions for further research”  “

This has been added

Round 2

Reviewer 3 Report

Suggested comments and corrections have been fully addressed.

No other issues raised, so paper can be published